# Generation of isogenic models of Angelman syndrome and Prader-Willi syndrome in CRISPR/Cas9-engineered human embryonic stem cells

Rachel B. Gilmore[1,2], Dea Gorka[1,2], Christopher E. Stoddard[1], Pooja Sonawane[1,2], Justin Cotney[1,3‡]*, Stormy J. Chamberlain[1,3‡]*

1 Department of Genetics and Genome Sciences, University of Connecticut School of Medicine, Farmington, CT, United States of America, 2 Graduate Program in Biomedical Science, Genetics and Developmental Biology, UConn Health, Farmington, CT, United States of America, 3 Institute for Systems Genomics, University of Connecticut, Storrs, CT, United States of America

☯ These authors contributed equally to this work.
‡ JC and SJC also contributed equally to this work.
* cotneyj@chop.edu (JC); stormy.chamberlain@roche.com (SJC)

**Data Availability Statement:** "All relevant data are within the paper and its Supporting Information

## Abstract

Angelman syndrome (AS) and Prader-Willi syndrome (PWS), two distinct neurodevelopmental disorders, result from loss of expression from imprinted genes in the chromosome 15q11-13 locus most commonly caused by a megabase-scale deletion on either the maternal or paternal allele, respectively. Each occurs at an approximate incidence of 1/15,000 to 1/30,000 live births and has a range of debilitating phenotypes. Patient-derived induced pluripotent stem cells (iPSCs) have been valuable tools to understand human-relevant gene regulation at this locus and have contributed to the development of therapeutic approaches for AS. Nonetheless, gaps remain in our understanding of how these deletions contribute to dysregulation and phenotypes of AS and PWS. Variability across cell lines due to donor differences, reprogramming methods, and genetic background make it challenging to fill these gaps in knowledge without substantially increasing the number of cell lines used in the analyses. Isogenic cell lines that differ only by the genetic mutation causing the disease can ease this burden without requiring such a large number of cell lines. Here, we describe the development of isogenic human embryonic stem cell (hESC) lines modeling the most common genetic subtypes of AS and PWS. These lines allow for a facile interrogation of allele-specific gene regulation at the chromosome 15q11-q13 locus. Additionally, these lines are an important resource to identify and test targeted therapeutic approaches for patients with AS and PWS.

## Introduction

Deletions of the maternal or paternal alleles of chromosome 15q11-q13, respectively, cause Angelman syndrome (AS [OMIM #105830]) and Prader-Willi syndrome (PWS [OMIM

files. Graphical representation of the deletions made in this work including specific break points generated can be explored on the UCSC Genome Browser in this session https://genome.ucsc.edu/s/rbgilmore/Megabase_Deletion_Lines_Figure_ClinVar".

**Funding:** National Institutes of Health (NIH) grants: T32HG010463 (R.B.G.), R35GM119465 (J.C.), R01 HD099975 (J.C. and S.J.C.), and R01HD094953 (S.J.C.) https://www.nih.gov/ The funders did not play any role in study design, data collection and analysis, decision to publish, or preparation of the manuscript.

**Competing interests:** S.J.C. is now an employee of F. Hoffmann-La Roche AG. This does not alter our adherence to PLOS ONE policies on sharing data and materials.

#176270]). Each occur at an approximate incidence of 1/15,000 to 1/30,000 live births [1–3]. Clinical features of AS include seizures, intellectual disability, absent speech, ataxia, and characteristic happy demeanor [4]. Other common features include microcephaly, abnormal EEG, sleep disturbances, strabismus, and, in cases where *OCA2* is deleted, hypopigmentation [5]. AS can be attributed to loss of function of *UBE3A* [6, 7]. Clinical features of PWS include neonatal hypotonia and failure-to-thrive during infancy, followed by hyperphagia and obesity; small stature, hands and feet; mild to moderate cognitive deficit and behavioral problems similar to obsessive–compulsive disorder [8–10]. While PWS is generally thought to be a contiguous gene disorder, recently described microdeletion cases encompassing just the *SNORD116* cluster highlight its crucial role in PWS pathophysiology [11]. Another rare neurodevelopmental disorder which shares some phenotypes with PWS is Schaaf-Yang syndrome (SYS), which is caused by mutations in *MAGEL2*, a protein-coding gene within the 15q11-q13 locus [12]. AS and PWS can be caused by a few different molecular mechanisms, but the most common is a large deletion, affecting ~70% of patients [13, 14]. The chromosome 15q11-q13 locus harbors intrachromosomal segmental duplications that can misalign during meiosis to generate these "common" large deletions within this chromosomal region [15].

Many of the genes in this region are governed by genomic imprinting, a phenomenon in which genes are expressed exclusively from one parental allele, rendering them functionally haploid. Deletion of single copies of the expressed alleles of these imprinted genes cause their full loss of function. Genomic imprinting at chromosome 15q11-q13 is established in the germline via differential methylation at the Prader-Willi Syndrome Imprinting Center (PWS-IC) [16–18]. The PWS-IC is methylated on the maternal allele and unmethylated on the paternal allele. This region on the unmethylated paternal allele serves as the canonical promoter for *SNRPN* transcript which is exclusively expressed from the paternally inherited allele. The *SNRPN* transcript is bi-cistronic, encoding for *SNURF* and *SNRPN* [19], and also codes a long non-coding RNA (lncRNA), *SNHG14* (reviewed by Ariyanfar & Good, 2022 [20]). *SNHG14* can be divided into two units, proximal and distal, based on its expression pattern. The proximal unit is broadly expressed across multiple tissue types and includes *SNURF-SNRPN*, *SNORD107*, *SNORD64*, *SNORD108*, *IPW*, *SNORD109A*, and *SNORD116*. The distal unit is exclusively expressed in neural cell types and includes *SNORD115*, *SNORD109B*, and *UBE3A-ATS* [21, 22]. The *UBE3A-ATS* portion of the transcript is responsible for silencing the paternal copy of *UBE3A* [23], thus expression of *UBE3A* occurs exclusively from the maternal allele in neurons. Many of the encoded RNAs are of the small nucleolar RNA (snoRNA) class, which are generally thought to be processed by exonucleolytic trimming from the introns of a host gene [24]. *SNORD116* and *SNORD115* are two clusters of snoRNAs, with 30 individual snoRNA copies and 48 individual snoRNA copies respectively. *SNORD116* can be further subdivided into three subgroups: Group I (*SNOG1*, *SNORD116-1* to *SNORD116-9*), Group II (*SNOG2*, *SNORD116-10* to *SNORD116-24*), and Group III (*SNOG3*, *SNORD116-25* to *SNORD116-30*) [21, 25]. Protein-coding genes *MKRN3*, *MAGEL2*, and *NDN*, are positioned upstream of the PWS-IC and are exclusively expressed from the paternally inherited allele.

The generation of patient-derived induced pluripotent stem cells (iPSCs) has led to increased understanding of human gene regulation at the chromosome 15q11-q13 locus [26–29]. However, variability in the genetic background and epigenetic reprogramming between different iPSC lines make it difficult to study the functional consequences of 15q imprinting disorders in neural cells. Here, we report the generation and characterization of isogenic chromosome 15q11-q13 megabase-scale deletions to model the most common genetic subtypes of AS and PWS. These models were built in the well-characterized and user-friendly H9 human embryonic stem cell (hESC) line to make use of the extensive publicly available data [30, 31] and robust neuronal differentiation [32, 33]. The use of isogenic cell lines provides a more rigorous

approach to investigate cellular deficits in disease models. These cell lines are well-suited for identifying quantitative molecular and physiological phenotypes, increasing confidence that observed differences between disease and control cells are due to the genetic disorders.

## Materials and methods

### hESC culture for editing and neuronal differentiation

hESC were maintained on mitotically inactivated mouse embryonic fibroblasts (MEFs) in feeding media which consists of sterile-filtered DMEM/F12 media (Gibco, #11330032) supplemented with 20% Knock Out Serum Replacement (Gibco, #10828028), 1X MEM Non-essential amino acids (Gibco, #11140050), 1mM L-glutamine (Gibco, #25030081) with 0.14% β-mercaptoethanol, and 8ng/mL bFGF (Gibco, #PHG0023). A humidified incubator with 5% $CO_2$ was used to maintain the cells at 37˚C. Stem cells were manually passaged by cutting and pasting colonies every 6 or 7 days using a 28-gauge needle. Stem cell media was replaced daily.

### hESC culture for hESC sample collection

To transition cells from feeder conditions to feeder-free conditions, cells were manually passaged by cutting and pasting colonies once confluent. After 5–7 days, any differentiation was manually removed before first passage. hESCs were maintained in mTeSR™ Plus media (STEMCELL Technologies, Catalog #100–0276) on Matrigel™ hESC-Qualified Matrix (Corning™, Catalog #354277) coated 6-well plates in a humidified atmosphere with 5% $CO_2$ at 37˚C. Feeding media was changed daily. Cells were passaged once 80–100% confluency was reached, approximately every 4–5 days. Briefly, media was removed from well(s), well(s) were gently rinsed with sterile PBS, sterile filtered 0.5 mM EDTA in PBS was added to well(s), and the plate was placed back into the incubator undisturbed for 2–5 minutes. After incubation, EDTA solution was gently aspirated from well(s), being careful to not disturb cells. Using a 2-mL serological pipette, 1 mL of media was added to well(s) while gently scraping bottom of well(s) to dislodge cells. 75–125 μL of cell suspension was added to a new well containing 2 mL of culture media supplemented with 10 μM ROCK inhibitor, Y-27632 2HCl (Tocris, #1254).

### Genome editing of hESCs

H9 ESCs were obtained from WiCell (WA09) and subsequently engineered with a megabase-scale deletion on either the paternal or maternal chromosome 15q allele. A similar editing and screening strategy has been previously described [34].

**Preparation.** A guide RNA targeting *GOLGA8* was designed using available guide RNA design tools (*GOLGA8* gRNA: `CTGGGTGTGAGGGCACGTGG`). The guide was cloned into the pSpCas9(BB)-2A-Puro (PX459) V2.0 plasmid (Addgene, #62988) via restriction digestion and ligation. Two days prior to planned genome editing, a 100mm dish of mitotically inactivated DR4 MEFs was prepared. A ~60–75% confluent well of hESCs was treated 24 hours prior to planned genome editing with 10μM ROCK inhibitor, Y-27632 2HCl (Tocris #1254).

**Nucleofection.** The day of editing, one ~75% confluent well of a 6-well plate of hESCs was treated with Accutase (Millipore Sigma, #SCR005) to release the cells from the plate. The cell suspension was singularized by pipetting and then pelleted. The media was removed from the cell pellet, and cells were resuspended according to the protocol provided for the P3 Primary Cell 4D-Nucleofector Kit (Lonza, V4XP-3024). Briefly, a mixture of 82μL nucleofector solution, 18μL nucleofection supplement, and ~5 μg of CRISPR plasmid was added to the pellet. The pellet was resuspended in the solution by pipetting gently three times using a P200 pipet. The cell suspension was transferred to the nucleofection cuvette and nucleofection was

performed on the 4D-Nucleofector (Lonza) on the program for hESC, P3 primary cell proto-col. After nucleofection, hESC suspension was immediately transferred to the 100mm dish plated with DR4 MEFs containing hESC feeding media supplemented with 10μM ROCK inhibitor using the transfer pipet included in the kit.

**Selection.** Feeding media was changed 24 hours following transfection (Day 1 post-trans-fection) and supplemented with 0.5–1 ng/μL puromycin and 10μM ROCK inhibitor. As the Cas9 cassette was not stably integrated into these cell lines, selection was continued for 48 hours total to select cells transiently expressing the vector containing the gRNA, Cas9 protein, and puromycin resistance. On Day 2, the media was changed and supplemented with fresh 0.5–1 ng/μL puromycin and 10μM ROCK inhibitor. On Day 3, the media was changed and supplemented with fresh 10μM ROCK inhibitor. Subsequent media changes occurred every other day, supplemented with fresh 10μM ROCK inhibitor. Once small colonies became visi-ble, media changes occurred daily with fresh media alone. After ~2 weeks, each colony was manually passaged into its own well of a 24-well plate coated with MEFs via cutting and past-ing. Feeding media in the 24-well plate was supplemented with 10μM ROCK inhibitor to encourage cell attachment. 48 hours after passaging cells, the feeding media was changed. Approximately 4 days after passaging to a 24-well plate, a few colonies from each well were iso-lated into PCR tube strips and pelleted for screening.

**Screening.** The TaqMan® Gene Expression Cells-to-CT™ Kit (Invitrogen™, #4399002) was used to screen clones following manufacturer's protocol. Briefly, media was removed from cell pellets in PCR tube strips and diluted DNase I lysis solution was added. A reverse tran-scription (RT) reaction was performed. Finally, a real-time PCR was run utilizing TaqMan™ Assays to measure expression of *UBE3A* (Hs00166580_m1) (ThermoFisher, #4331182) versus *GAPDH* (ThermoFisher, #4352934E) in technical duplicates or triplicates at a total reaction volume of 20μL. Clones that were found to have ~50% reduction in *UBE3A* compared to wild type controls were further expanded and subjected to confirmatory testing. A previously described iPSC line derived from an AS patient (ASdel1-0) [26] was included in the assay as a control line for half *UBE3A* expression.

**Confirmatory testing.** After manually passing clones for expansion and verifying attach-ment of colonies in new wells, the remainder of cell colonies were scraped from old wells and pelleted in microcentrifuge tubes. Genomic DNA (gDNA) was extracted from clones using a homemade lysis buffer containing 1% sodium dodecyl sulfate (SDS), 75mM NaCl, 25mM EDTA, and 200μg/mL Proteinase K in UltraPure™ DNase/RNase-Free Distilled Water (Ther-moFisher, #10977015). Briefly, media was removed from each cell pellet and 250uL of the lysis buffer was added. Tubes were incubated at 60°C overnight. The following day, 85μL of warm 6M (supersaturated) NaCl was added, followed by the addition of 335μL of chloroform. The tubes were capped and then inverted for approximately one minute. Tubes were centrifuged at 9,000 rcf for 10 minutes at room temperature. The top aqueous layer (~335μL) was removed and transferred to a new tube to which an equal amount of 100% isopropanol was added. The tubes were capped and mixed thoroughly by inversion. Tubes were incubated at -20°C for ~10 minutes. Next, the tubes were centrifuged at max speed (~18,000 rcf) for 20 minutes at 4°C. The supernatant was removed, and the pellet was washed with ~600μL of 70% ethanol. Ethanol was removed carefully from the pellet and the tubes were left open so the remainder of the eth-anol could evaporate before the pellets were resuspended in 30μL of 10mM Tris (pH 8). Taq-Man™ Copy Number Assays comparing UBE3A (Hs01665678_cn)(ThermoFisher, #4400291) to the TaqMan Copy Number Reference Assay for human RNase P (ThermoFisher, #4403326) was performed following manufacturer's protocol (Applied Biosystems™, Publication Number #4397425) to confirm which clones had lost a copy of *UBE3A*. The wild type H9 line was used as the calibrator sample with two copies of *UBE3A*. The same AS1-0 iPSC line used as a control

in our screening assay was also included as a control in this assay as it only has one copy of *UBE3A*. Analysis was conducted using the CopyCaller (v2.1) software (Applied Biosystems®). Further confirmation of which allele was deleted in clones with only one *UBE3A* copy was performed by utilizing the EpiTect Methyl II DNA Restriction Kit (QIAGEN, #335452) to measure methylation at the PWS-IC (*SNRPN*) following manufacturer's protocols. The approximate size of the deletions was determined by CytoSNP array (Illumina, CytoSNP-850K v1.2) through the University of Connecticut Chromosome Core. Clones were further screened for integration of the targeting plasmids via PCR. Three amplicons were designed to test for insertion of the guide RNA sequence, the CMV promoter, and the Cas9 gene. Primers used for each amplicon are listed in S1 Table. Clones with confirmed deletions were expanded, banked down, and subsequently characterized via gene expression arrays in stem cells and neurons.

## Neuronal differentiation and maintenance

Neuronal differentiation was performed according to established monolayer differentiation protocols with minor modifications [35–38]. Approximately 1–3 days after passaging hESCs, neuronal differentiation was started (Day 0) by switching feeding media to N2B27 neural induction media supplemented with 500 ng/µL noggin (R&D Systems, #3344-NG). N2B27 neural induction media consisted of Neurobasal™ Medium (Gibco, #21103049), 1X serum free B-27™ Supplement (Gibco, #17504044), 1% N2 supplement, 1% insulin-transferrin-selenium (Gibco, #51300044), 2mM L-glutamine (Gibco, #25030081), and 1% penicillin-streptomycin (Gibco, #15140122). The media was changed every other day for 10 days and supplemented with fresh 500 ng/µL noggin on Days 2, 4, 6, and 8. Between Days 14–17, neuronal rosettes were passaged as small clusters in either a 1:1 or 1:2 ratio using the StemPro™ EZPassage™ Disposable Stem Cell Passaging Tool (Gibco, #23181010) or via hand-picking. Rosettes were plated on poly-D-lysine- (PDL-)(Millipore Sigma, #P0899) and laminin-coated (Gibco, # 23017015) 6-well plates. Fifty percent media replacement was carried out every other day until neural progenitor cells (NPCs) were dense enough for replating. At ~3 weeks, Accutase was used to release the cells from the plate. The cell suspension was singularized by pipetting and then pelleted. The media was removed from the cell pellet, NPCs were resuspended, and replated at a high density onto poly-D-lysine/laminin-coated 6-well plates into N2B27 media containing 10µM ROCK inhibitor. Fifty percent media replacement was carried out every other day. After approximately five weeks of neural differentiation NPCs were dissociated again using Accutase, counted using a hemocytometer, and plated on PDL/laminin-coated 6-well plates with at a density of 150,000–300,000 cells/well in neural differentiation medium (NDM). NDM consisted of Neurobasal™ Medium, 1X serum free B-27™ Supplement, 1X MEM Non-essential amino acids, 2mM L-glutamine, 10ng/mL brain-derived neurotrophic factor (BDNF)(Peprotech, #450–02), and 10ng/mL glial-derived neurotrophic factor (GDNF)(Peprotech, #450–10), 200µM ascorbic acid (Millipore Sigma, #A4544), and 1µM adenosine 3',5'-cyclic monophosphate (cAMP)(Millipore Sigma, #A9501). To aid in cell attachment, 10µM ROCK inhibitor was added to the NDM during initial plating. Cells were maintained with no antibiotics on NDM, and 50% media replacement was carried out twice per week. Gene expression assays were conducted on neuronal cultures that were at least 10 weeks old.

## Gene expression arrays

**RNA extraction.** Confluent hESCs or mature hESC-derived neurons were collected and gently pelleted at 4˚C. Supernatant media was removed from pellet. Pellet was flash frozen in liquid nitrogen and stored at -80˚C until all samples were collected. RNA was harvested using the miRNeasy® Mini Kit (QIAGEN, #1038703) following manufacturer's protocol with

minor modifications. The work surface, pipettes, and centrifuge rotors were treated with RNAse Away (Life Technologies, #10328011) prior to beginning extraction. Pellets were transferred from storage at -80˚C to ice. Samples were homogenized in 700 µL QIAzol by pipetting and brief vortexing. Cell lysate was applied to QIAshredder columns (QIAGEN, #1011711). Samples were incubated at room temperature for 5 minutes. Following incubation, 140 µL of chloroform was added to the homogenate and shaken vigorously for 15 seconds. Samples were incubated at room temperature for 2–3 minutes and then centrifuged for 15 minutes at 12,000xg at 4˚C. Approximately 400 µL of the aqueous phase was transferred to a new 1.5-mL microcentrifuge tube. A second chloroform extraction was performed by adding an equal volume of chloroform to the aqueous phase and shaking vigorously for 15 seconds. The samples were centrifuged for another 15 minutes at 12,000xg at 4˚C, and the aqueous phase (~350 µL) was transferred to a new 1.5-mL microcentrifuge tube to which 1.5 volumes of 100% ethanol was added. The contents of the tube were mixed by pipetting and applied to the RNeasy spin column, following manufacturer's instructions for on-column DNase treatment using RNase-Free DNase Set (QIAGEN, #79254) and the addition of a second wash with Buffer RPE. RNA was eluted in RNase-Free water and stored at -80˚C until RT-qPCR processing. All samples were performed in biological triplicate, one passage apart for hESCs and separate differentiations for neurons.

**RT-qPCR.** The High-capacity cDNA Reverse Transcription Kit (Applied Biosystems™, #4368814) was used to generate cDNA from DNase-treated RNA following manufacturer's protocol. Custom TaqMan Gene Expression Assay Plates (Thermo Fisher Scientific, #4391528) were used with TaqMan™ Gene Expression Master Mix (Applied Biosystems™, #4369016) to measure gene expression following manufacturer's protocol for a total reaction volume of 20 µL. All samples were performed in technical duplicate or triplicate for each gene measured. The list of TaqMan probe RT-qPCR assays used in the study are provided in S1 Table.

## Data analysis

For analysis of qPCR data, first the Ct values for technical replicates were tested for outliers using the Grubbs test (S2 Table). Ct values that fell beyond 99% of the range of the characterized distribution were considered outliers and removes, as has been previously described [39]. The mean Ct value of technical replicates for each gene were normalized to the Ct value for housekeeping gene *GAPDH*. Relative expression was quantified as $2^{-\Delta\Delta Ct}$ relative to the wild type H9 sample. Data are presented as the mean relative expression, plus or minus the relative min/max as calculated by error propagation defined by Ahmed and Kim [40]. Error bars represent relative min/max calculated with error propagation. Statistical analysis was performed using a one-way ANOVA followed by Dunnett's test. **** $p<0.0001$, *** $p<0.001$, ** $p<0.01$, * $p<0.05$. Calculations were performed in Excel and can be found in S2 Table. Bar graph generation was performed using R (v4.2.1) with ggplot2 (v3.4.0) and tidyverse (v1.3.2). Code for plotting can be found at https://github.com/rachelgilmore/qPCRanalysis.

## Immunocytochemistry

hESCs were plated on Matrigel-coated acid-treated coverslips. hESC-derived neurons were differentiated as above and plated for terminal differentiation on PDL and laminin-coated acid-treated coverslips. Once hESCs reached ~50% confluency or neurons reached 10 weeks of maturity, wells were rinsed twice with PBS. Cells were fixed at room temperature with 4% paraformaldehyde for 10 min and then permeabilized using PBS plus 0.5% Triton X 100 (PBS-T) for 5 min at room temperature. Following permeabilization, samples were blocked in

0.1% PBS-T containing 2% bovine serum albumin and 5% normal goat serum. Samples were incubated in blocking buffer containing primary antibodies overnight at 4˚C in a humidity chamber. The next day the coverslips were washed with three times with 0.1% Triton in PBS for 10 minutes each. Samples were then incubated in blocking buffer containing secondary antibodies for 1 hour at room temperature in the dark. All remaining steps occurred in the dark. The coverslips were washed three times with 0.1% Triton in PBS for 10 minutes each. Coverslips were mounted with ProLong™ Gold Antifade Mountant with DNA Stain DAPI (Invitrogen, Cat# P36941) and allowed to set for 24 hours at room temperature prior to imaging. The following primary antibodies were used: mouse anti-Oct4 (1:200; Abcam ab184665), rat anti-NeuN (1:1000; Abcam ab279279), and rabbit anti-MAP2 (1:800, Abcam ab32454). The following secondary antibodies were used: goat anti-rabbit Alexa Fluor 488 (1:400; Invitrogen A11008), goat anti-rat Alexa Fluor 594 (1:400; Invitrogen A11007), and goat anti-mouse Alexa Fluor 647 (1:400, Invitrogen A21235). Images were acquired using a 20X or 63X objective on a Zeiss Axio Observer Z1 microscope. Representative images were chosen, and image adjustment was performed in ImageJ. In all photos, only the color balance was adjusted. Images were assembled using Adobe Illustrator.

## Results

To generate isogenic models of AS and PWS, we sought to recapitulate deletions frequently present in AS or PWS patients. Examination of deletions deposited in ClinVar and DECI-PHER [41, 42] revealed breakpoint hotspots that coincide with repeats of *GOLGA8* (Fig 1). This repeated sequence described as contributing significantly to substantial instability at this locus [43, 44]. Therefore we pursued targeting these segmental duplications, similar to the approach used to eliminate the Y-chromosome or trisomic chromosome 21 [45, 46]. We previously used a similar CRISPR/Cas9 strategy, with guide RNAs (gRNAs) targeting *GOLGA8* and other repetitive sequences in chromosome 15, to evict an extra chromosome and generate an isogenic model for Duplication 15q Syndrome (Dup15q, [OMIM #608636]) [34]. Others have created 15q13.3 microdeletions leveraging this approach [47].

Building on the concepts utilized in these previous studies, we began by nucleofecting H9 hESCs with a plasmid encoding CRISPR/Cas9 and a single gRNA targeting *GOLGA8* repeats on chromosome 15q (Fig 1) (Methods). This gRNA is predicted to target multiple sites within chromosome 15q but is not predicted to target elsewhere in the genome. We screened clones surviving transient puromycin selection, which eliminated cells that did not receive the Cas9/gRNA plasmid, for expression of *UBE3A* with a TaqMan-based assay (Methods). As stem cells bi-allelically express *UBE3A*, cell lines harboring AS and PWS related deletions should therefore express approximately half as much *UBE3A* as the parent H9 line (Fig 2A). This screening method provided us with a high-throughput way to screen 126 clones from 4 separate transfections (S1 Fig and S3 Table). Three clones with reduced *UBE3A* expression comparable to an Angelman iPSC line (ASdel1-0) were expanded and subject to confirmatory testing. While our initial screen utilized cDNA and relative expression of *UBE3A*, we confirmed deletions by determining the copy number of UBE3A in genomic DNA (gDNA) extracted from our edited clones. We compared edited clones to wild type H9 samples harboring two UBE3A copies and the ASdel1-0 sample harboring only one UBE3A copy. All three clones were predicted to contain a single UBE3A copy (Methods) (Fig 2B and S4 Table). To determine the parent-of-origin of the deletions in our three cell lines, we subjected gDNA isolated from them to methylation analysis at the Prader-Willi Syndrome Imprinting Center (PWS-IC, *SNRPN*)(Methods). A wild type cell line will show ~50% methylation at *SNRPN*, as the paternal allele is unmethylated and the maternal allele is methylated. Previous analysis of patient-derived AS and PWS lines

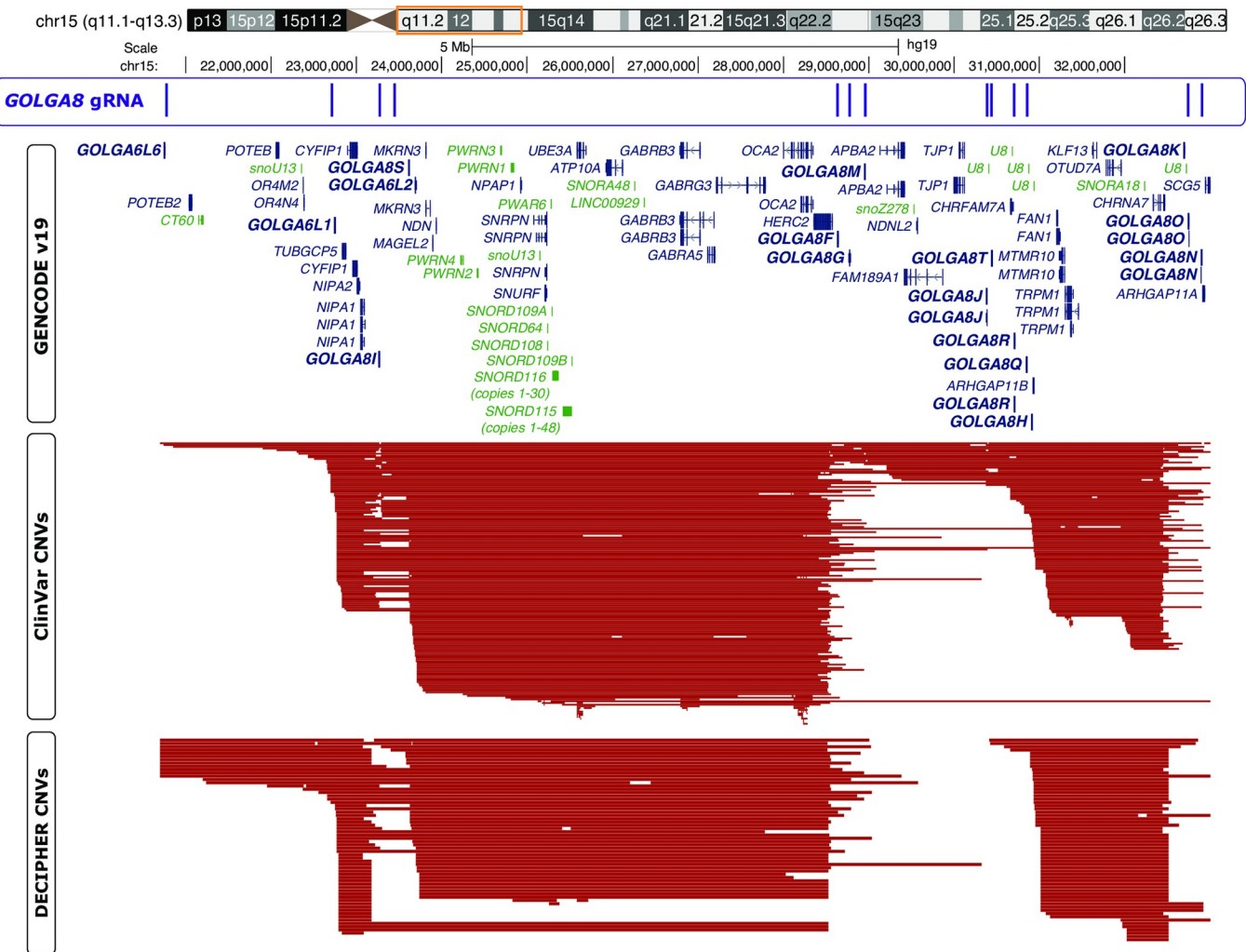

**Fig 1. UCSC genome browser shot of the chromosome 15q locus.** Chromosome ideogram with orange box around region displayed below. First track displays *GOLGA8* CRISPR gRNA binding sites. Second track displays GENCODEv19 gene annotations; genes in dark blue are protein coding, genes in green are non-coding, arrows indicate direction of gene transcription. Some gene isoforms have been removed for clarity. GOLGA genes are shown in bold. Third track displays ClinVar copy number variants (CNVs); only deletions with pathogenic or likely pathogenic annotations are shown. Fourth track shows DECIPHER CNVs, only deletions with pathogenic or likely pathogenic annotations are shown.

showed that, as expected, PWS deletion lines only have a methylated maternal allele and are ~100% methylated, and AS deletion lines only have an unmethylated paternal allele lacking methylation at this CpG island [26]. The methylation analysis indicated that two of the three clones with reduced *UBE3A* expression and copy number exhibited a primarily unmethylated PWS-IC resembling AS (H9Δmat15q_1 and H9Δmat15q_2) and one clone exhibited a primarily methylated PWS-IC resembling PWS (H9Δpat15q) (Fig 2C and S5 Table). To determine if the plasmid used to target the deletions might have stably integrated into the genome we screen each clone with three different amplicons. We found no evidence of guide RNA or functional Cas9 gene integration into the genomes of any of the chosen clones (S2 Fig). To determine the approximate size of the deletion, clones were further characterized by a CytoSNP analysis. This analysis revealed an ~5.8Mb deletion in the H9Δmat15q_1 line, an ~8Mb deletion in the H9Δmat15q_2 line, and two deletions totaling ~7Mb in the H9Δpat15q line (S3 Fig). The coordinates of the deletions returned from CytoSNP analysis were displayed

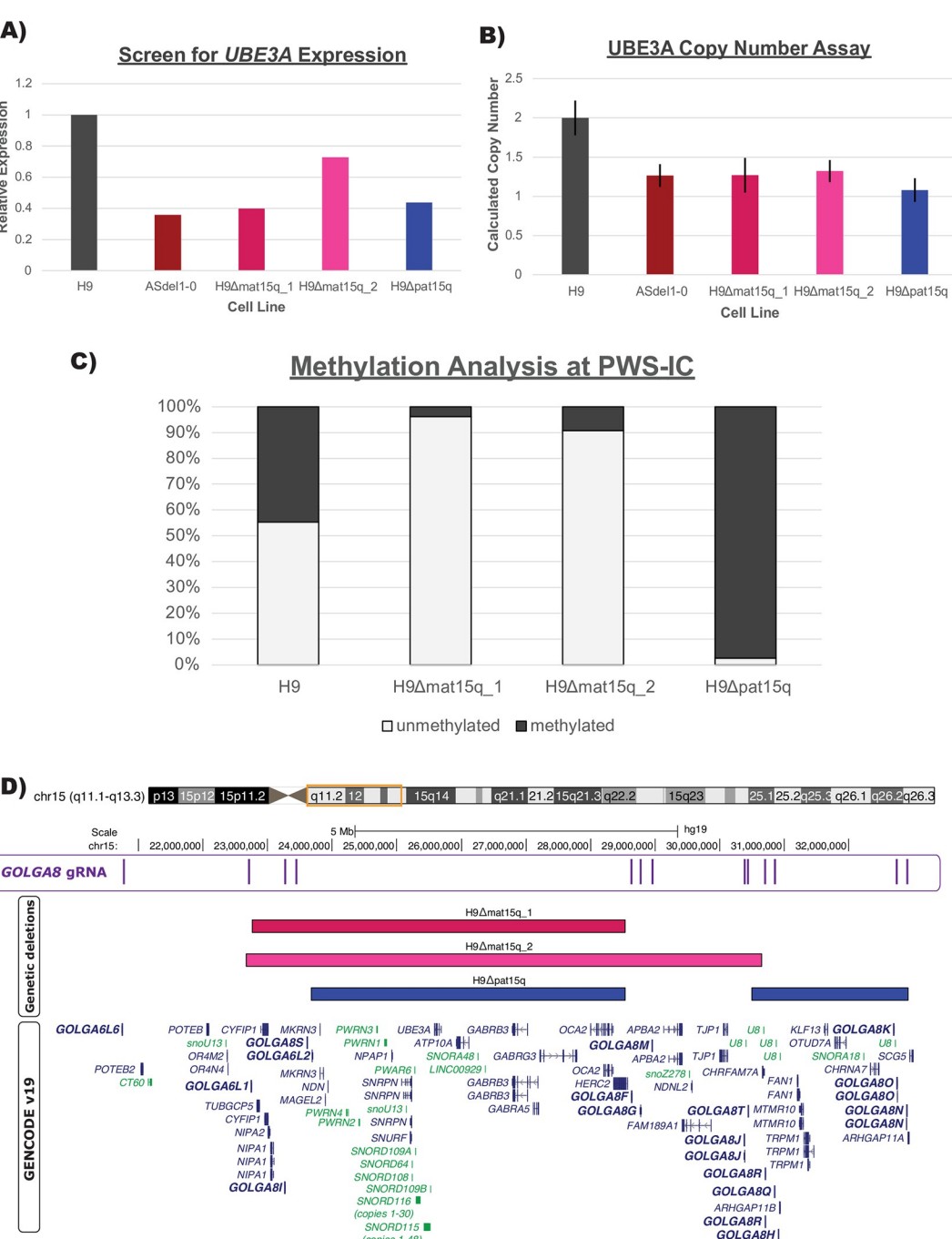

**Fig 2. Edited clone screening and confirmatory testing.** A) Bar plot of relative *UBE3A* expression of edited clones compared to wild type (H9) ESCs and Angelman syndrome (ASdel1-0) iPSCs. B) Bar plot of calculated UBE3A copy number of edited clones. Wild type (H9) was used as the calibrator sample and set to 2 copies. Error bars represent calculated copy number range. Copy number and calculated copy number was determined by CopyCaller v2.1 software. C) Stacked bar plot representing methylation status at the PWS-IC/*SNRPN* locus. Light gray bar represents unmethylated DNA, dark gray bar represents methylated DNA. D) UCSC Genome Browser shot of the chromosome 15q locus. First track displays *GOLGA8* CRISPR gRNA binding sites. Second, third, and fourth tracks show deletion contained within each cell line, as determined by CytoSNP. Fifth track displays GENCODEv19 gene annotations; genes in dark blue are protein coding, genes in green are non-coding, arrows indicate direction of gene transcription. Some gene isoforms have been removed for clarity. GOLGA genes are shown in bold.

as bedtracks in the UCSC genome browser to visually display the size of the deletions and which genes may be impacted (Fig 2D). These deletions coincided well with those observed in patients (S4 Fig), further supporting *GOLGA8*-driven instability as genetic mechanism leading to AS and PWS.

We next sought to determine the pluripotent potential of these lines and to characterize their gene expression profile in the chromosome 15q locus as ESCs (Methods)(S2 Table). There were very few significant differences in pluripotency genes of the edited cell lines compared to the parental H9 line (Fig 3A). Additionally, we performed immunocytochemistry for Oct4 protein, which was robustly expressed in the line analyzed (S5 Fig). In lines with maternal deletions, imprinted 15q genes showed similar expression to the parental H9 line as expected (Fig 3B). In the paternal deletion line, imprinted 15q genes that are expressed exclusively from the paternal allele showed very little expression compared to the parental H9 line (Fig 3B). This is expected, as these imprinted genes are silenced on the intact maternal allele. This data supported our characterization of which allele was deleted in each cell line. Most bi-allelically expressed 15q genes within the deletion breakpoints showed approximately half of the expression of the parental H9 line (Fig 3C) excluding *GABRG3* which was inconclusive (S5 Fig).

As differentiation into the neuronal lineage is important for studying these disorders and is crucial for verifying the imprinting status of *UBE3A* in neurons, we performed neuronal differentiation of all lines (Methods). All lines successfully differentiated into neurons (Fig 4A and S6 Fig). As expected, expression of neuronal genes was comparable to the wild type H9 parental line (Fig 4B and S6 Fig). Along with similar gross morphology and comparable gene expression across samples, immunocytochemistry revealed expression of neuronal proteins MAP2 and NeuN (S6 Fig). Very little expression from imprinted 15q genes was observed for the paternal deletion line (Fig 4C). For maternal deletion lines, most paternally imprinted genes are expressed comparable to that of the parental H9 line as expected, except *UBE3A*, which was significantly reduced (Fig 4C and S6 Fig). We considered this proof-of-concept that these isogenic AS-like cell line models are capable of imprinting *UBE3A* following neuronal differentiation. Bi-allelic 15q gene expression maintained a similar expression profile to that observed in ESCs, with most genes contained within the deletion showing approximately half expression compared to the parental H9 line (Fig 4D and S6 Fig). Five of the seven pluripotency genes measured show decreased expression in neurons compared to wild type ESCs, while neuronal genes show robustly increased expression compared to wild type ESCs (S7 Fig). Given that these cell lines readily differentiate into neurons, we anticipate they will provide great utility for understanding the specific role these deletions play in neurodevelopmental phenotypes of AS and PWS.

## Discussion

While the genetic perturbations contributing to AS and PWS have been known for many years, the effect those anomalies have on the chromosome 15q locus and the genome as a whole remains unclear. Mouse models have provided key understandings about the facets of gene regulation conserved between the two species, but disease features such as the neuron-specific regulation of *UBE3A* imprinted expression and protein targets of UBE3A appear to be unique to humans [22, 48–50]. Available iPSC models have been powerful tools to study these disorders as well. However, comparison of quantitative molecular phenotypes via multiomics approaches and functional studies of iPSC-derived neurons have been hampered by the variability between iPSC lines. While a prior study engineered a type I deletion modeling PWS into iPSCs through use of multiple CRISPR gRNAs [51], here we have described the first isogenic cell line pairs modeling the most common genetic subtypes of both AS and PWS

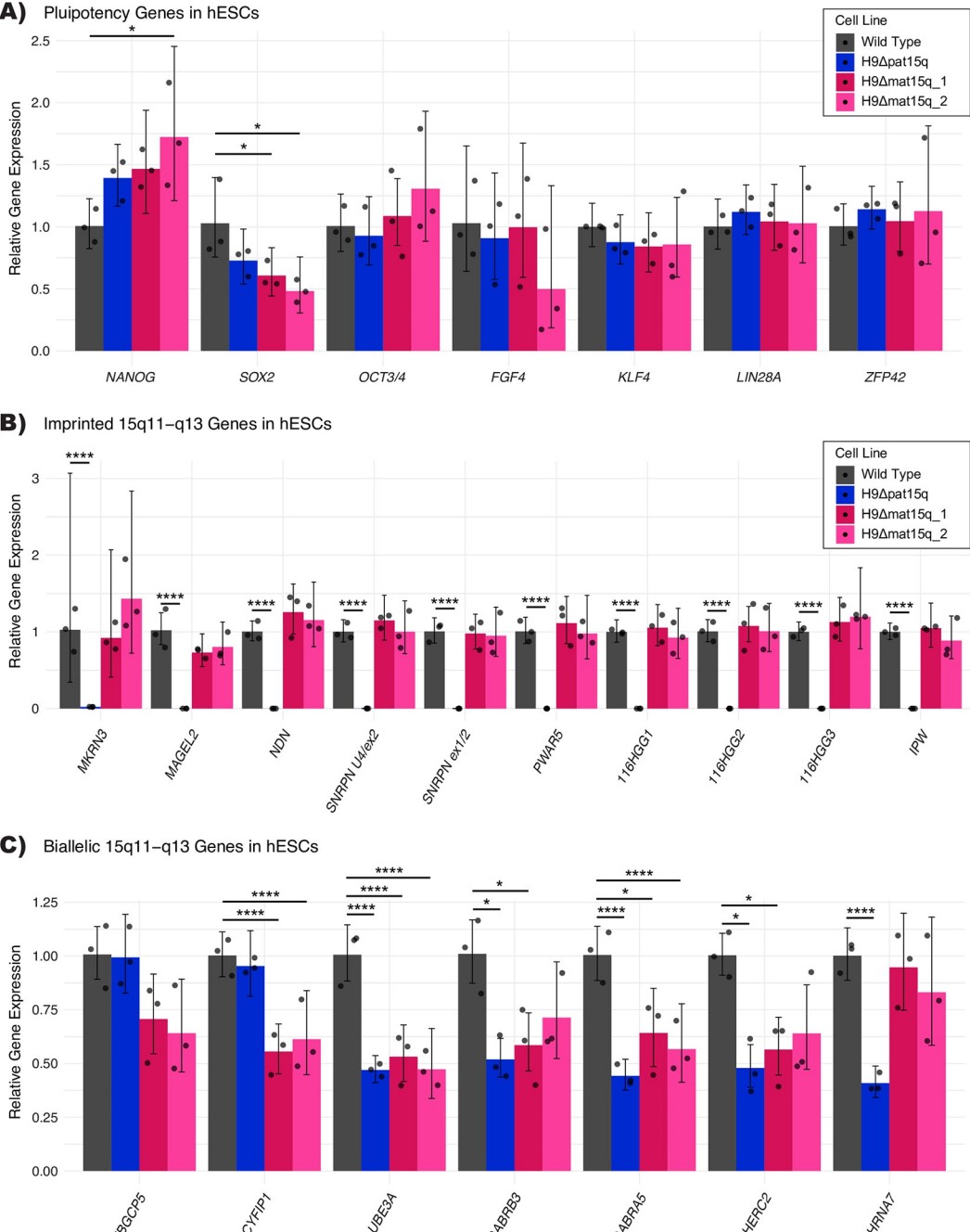

**Fig 3. hESC characterization.** qPCR analysis of A) pluripotency markers, B) imprinted genes, and C) bi-allelically expressed genes in the 15q locus in edited clones as ESCs (n = 3 biological replicates). RNA expression is presented relative to the parental H9 ESC line. Error bars represent relative min/max calculated with error propagation. Statistical analysis was performed using a one-way ANOVA followed by Dunnett's test. Significance is reported as result of Dunnett's test. **** p<0.0001, *** p<0.001, ** p<0.01, * p<0.05.

utilizing a single CRISPR gRNA in a well-characterized H9 ESC line. We reasoned that we may be able to mimic megabase-scale deletions found in patients by targeting chromosome 15q-specific *GOLGA8* repeats. We took advantage of the bi-allelic expression of *UBE3A*, a

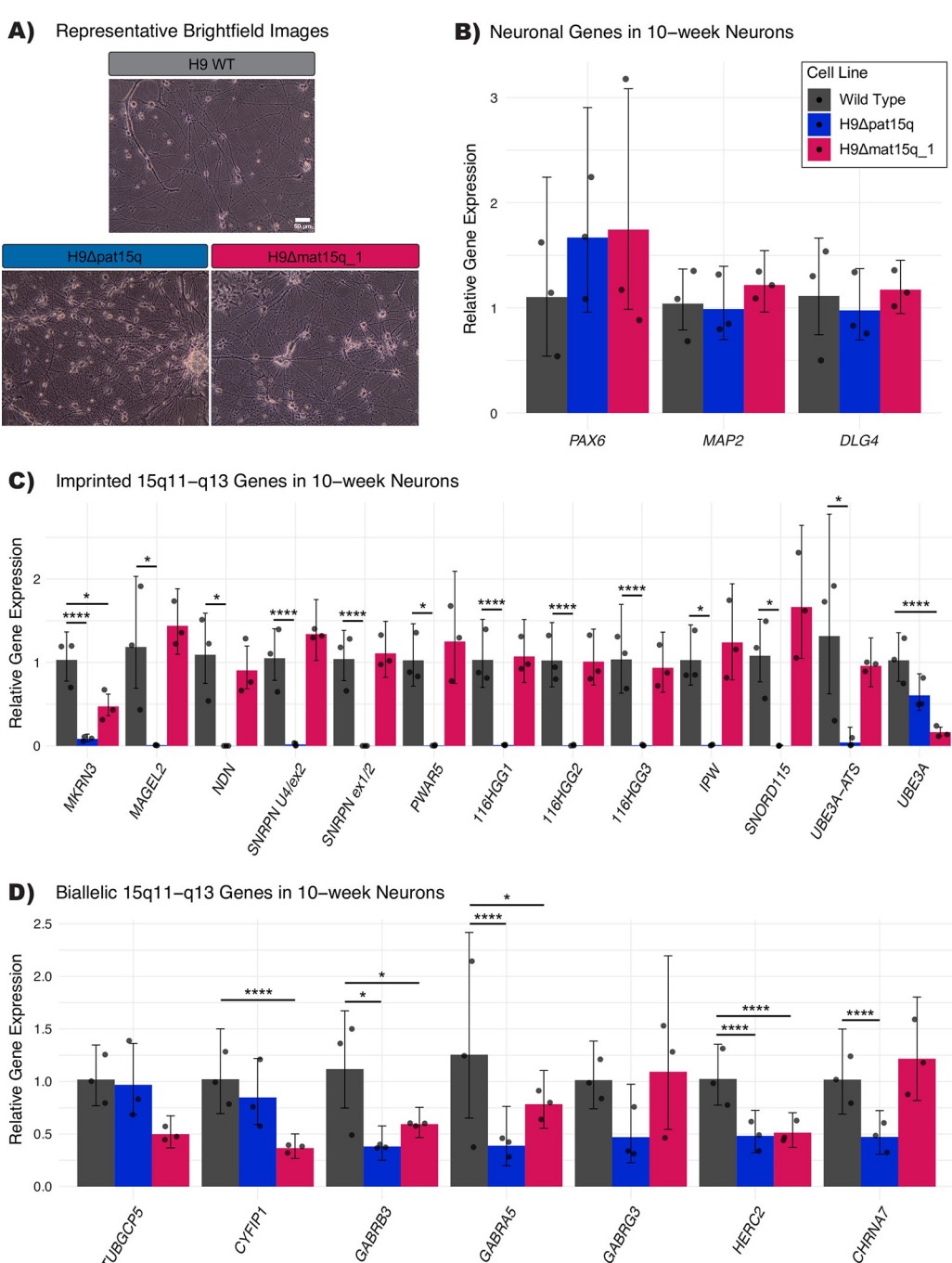

**Fig 4. Mature neuron characterization.** A) Representative brightfield images of neurons at 20X magnification. Scale bar equals 50 μm. B-D) qPCR analysis of B) neuronal genes C) imprinted genes in the 15q locus and D) bi-allelically expressed genes in the 15q locus in edited clones as mature 10-week neurons (n = 3 biological replicates). RNA expression is presented relative to the parental H9 line. Error bars represent relative min/max calculated with error propagation. Statistical analysis was performed using a one-way ANOVA followed by Dunnett's test. **** $p<0.0001$, *** $p<0.001$, ** $p<0.01$, * $p<0.05$.

gene included in the deleted region, in hESCs to rapidly screen for edited clones with reduced expression (Fig 2A and S1 Fig). Our rationale was that if either the maternal or paternal chromosome 15q allele was deleted in this region, we would observe approximately half the *UBE3A*

expression compared to a wild type control. We subjected the clones with reduced *UBE3A* expression to a more rigorous confirmatory testing utilizing a copy number assay to determine the number of *UBE3A* copies present in gDNA extracted from each clone (Fig 2B).

As the parent-of-origin of the deletion matters, we utilized differential methylation at the PWS-IC/*SNRPN* to determine which allele was deleted. The differential methylation at this site has been characterized previously in patient-derived iPSCs [26]. We hypothesized our isogenic models would have comparable methylation signatures to iPSC models if they harbored similar deletions, which was what we observed (Fig 2C). Confident we created deletions on either the paternal or maternal allele, we employed a CytoSNP array to determine the approximate size of the deletion. This assay leveraged microarray technology to detect copy-neutral loss of heterozygosity (LOH), absence of heterozygosity (AOH), and copy number variation (CNV) via gains or losses. While off-target editing is sometimes a concern with CRIPSR/Cas9 editing, CytoSNP analysis did not detect any copy number changes or structural rearrangements aside from the deletions on chromosome 15. Any copy number changes outside of chromosome 15 in the edited hESCs were also present in the parental H9 cell line previously characterized by the lab, supporting the isogenic nature of these cell lines (S3 Fig).

Having determined the parent-of-origin and approximate size of the deletions, we wanted to ensure these edited cell lines were still pluripotent. While the expression profile of the pluripotency markers in edited cell lines did not match exactly to the wild type controls (Fig 3A), this could be caused by differences in the quality of the cultures at the time of collection. The pluripotent potential of the edited cell lines was supported by their ability to differentiate successfully into a neuronal lineage (Fig 4A). We also wanted to examine gene expression within the chromosome 15q locus. We would expect the imprinted gene expression in maternal deletion lines to be consistent with the parental wild type line, as the imprinted and unexpressed copy of each gene was deleted. We observe no significant differences in imprinted 15q gene expression in maternal deletion ESC lines, however we do notice some significant differences in imprinted genes in the H9Δmat15q_2 line in 10-week neurons (Fig 4C). We believe these differences are driven by one sample which was analyzed via qPCR at a lower concentration. We decided to include this sample in the analysis so there were three biological replicates, though further replication should be performed in future studies to determine if this variation is truly biological in nature. In contrast, we would anticipate little to no expression of the paternally expressed imprinted genes in the paternal deletion line, as the expressed copy of each gene was deleted, which was exactly what we observed (Figs 3B and 4C).

The bi-allelically expressed genes included within the deleted region generally showed approximately half of the expression of the parental H9 line as expected, though the analysis of H9Δmat15q_2 neurons was again complicated by the lower concentration sample. While *GABRG3* was included in the deletion generated for all lines, its expression was not reliably measured within our system. Whether this is biological or technical is inconclusive. While *UBE3A* is bi-allelically expressed in ESCs, the paternal *UBE3A* copy undergoes silencing in neurons [21–23, 26]. We showed that both maternal deletion demonstrated evidence of *UBE3A* imprinting (Fig 4C and S6 Fig). Successful neuron differentiation for all lines was shown through typical neuron morphology (Fig 4A and S6 Fig), increase in neuronal gene expression (S7 Fig), neuronal protein expression (S6 Fig), and decrease in expression of pluripotency genes (S7 Fig). *SOX2* and *KLF4*, two genes canonically referred to as pluripotency factors, did not decrease in expression. This could be due to their role in neural stem cell maintenance [52] and neurogenesis [53]. Further functional studies of these hESC-derived neurons could determine if they display similar deficits to those found in iPSC-derived neurons [54].

These isogenic cell lines provide a powerful resource to carefully discern cellular and molecular phenotypes between disease and wild type states for these large chromosomal deletions. We posit that the use of these isogenic pairs will lead to more robust and reproducible results, especially when combined with additional isogenic pairs and/or patient-derived iPSC lines. Additionally, these resources may open the door for the discovery of novel, more specific therapeutic approaches for AS and PWS patients. Lastly, this work adds to the current literature supporting the utility of CRISPR/Cas9 editing to eliminate large regions of the genome and its application to model disorders with large copy number variants.

## Supporting information

**S1 Fig. Compilation of screening data from *UBE3A* expression from all clones from all transfections.** Expression of *UBE3A* is set to a relative value of 1 for the wild type parental H9 line. As later determined through confirmatory testing, H9Δmat15q_1 is designated as Tfx1_Clone13, H9Δmat15q_2 is designated as Tfx4_Clone22, H9Δpat15q is designated as Tfx4_Clone2.
(PDF)

**S2 Fig. Screen for integration of PX459 GOLGA8 guide RNA plasmid.** Plasmid map of PX459 containing guide RNA targeting GOLGA8 repeat sequences. Primers for individual amplicons are indicated on the map. The plasmid was used as a positive control for PCR reactions 1 through 3. An additional amplicon for the *TBX5* promoter was used as a positive control for genomic DNA for each PCR reaction (number 4). For each of the lines, including the parental H9 line, the *TBX5* amplicon amplified. The PX459 amplicons were detected from purified plasmid DNA but failed to amplify from any genomic DNA sample from edited clones.
(PDF)

**S3 Fig. Reports from CytoSNP array.** H9Δmat15q_1 is designated as H9 AS #13, H9Δmat15q_2 is designated as H9 AS #22, and H9Δpat15q is designated as H9 PWS #2.
(PDF)

**S4 Fig. UCSC genome browser shot of the chromosome 15q locus.** First track displays *GOLGA8* CRISPR gRNA binding sites. Second, third, and fourth tracks show deletion contained within each cell line, as determined by CytoSNP. Fifth track displays GENCODEv19 gene annotations; genes in dark blue are protein coding, genes in green are non-coding, arrows indicate direction of gene transcription. Sixth track displays short (<50 bp) variants found in the ClinVar database as a density plot. Seventh track displays ClinVar copy number variants (CNVs); for clarity, only deletions with pathogenic or likely pathogenic annotations are shown. Eighth track shows DECIPHER CNVs, only deletions with pathogenic or likely pathogenic annotations are shown. Ninth track shows conservation scores. Tenth track shows alignment of sequences to other vertebrates.
(PDF)

**S5 Fig. Further characterization of hESCs.** A) Immunocytochemistry for Oct4 with nuclear marker DAPI. Images taken at 20X and 63X. Scale bars equal 50 μm and 25 μm, respectively. B) qPCR analysis of GABRG3 in hESCs (n = 3 biological replicates). RNA expression is presented relative to the parental H9 line. Error bars represent relative min/max calculated with error propagation. Statistical analysis was performed using a one-way ANOVA followed by Dunnett's test. Significance is reported as the results of Dunnett's test. * = p < 0.05.
(PDF)

**S6 Fig. Characterization of H9Δmat15q_2 neurons.** A) Representative brightfield images of neurons at 20X magnification. Scale bar equals 50 μm. B) qPCR analysis of neuronal genes. C) Immunocytochemistry for neuronal proteins MAP2 and NeuN. Images taken at 20X and 63X. Scale bars equal 50 μm and 25 μm, respectively. D-E) qPCR analysis of D) imprinted genes and E) bi-allelically expressed genes in the 15q locus. For all qPCR data presented, n = 3 biological replicates. RNA expression is presented relative to the parental H9 line. Error bars represent relative min/max calculated with error propagation. Statistical analysis was performed using a t-Test for two samples assuming equal variances. Significance is reported as the results of the two-tailed test. ** = $p < 0.01$, * = $p < 0.05$.
(PDF)

**S7 Fig. Comparative analysis of neurons and ESCs.** qPCR analysis of A) pluripotency genes and B) neuronal genes in mature 10-week neurons (n = 3 biological replicates). RNA expression is presented relative to the parental H9 line as ESCs. Error bars represent relative min/max calculated with error propagation. Statistical analysis was performed using a one-way ANOVA followed by Dunnett's test. Significance is reported as the results of Dunnett's test. **** $p < 0.0001$, *** $p < 0.001$, ** $p < 0.01$, * $p < 0.05$.
(PDF)

**S1 Table. TaqMan probes used in qPCR and primers used for integration screening.** Gene and probe name for each ThermoFisher TaqMan assay used. Primer names and sequence of each primer used for PCR performed in S2 Fig.
(XLSX)

**S2 Table. qPCR data and analysis.** H9Δmat15q_1 is designated as AS-13, H9Δmat15q_2 is designated as AS-22, and H9Δpat15q is designated as PWS-2. Sheet titled "RT Set Up" details the experimental set up for the reverse transcriptase reaction to convert RNA samples to cDNA. Sheet titled "qPCR Set Up" details the calculations for performing qPCR experiments. Sheet titled "qPCR Data" contains all raw data from qPCR instrument. Sheet titled "Grubbs Test" contains calculations for outlier testing using Grubbs test. Sheet titled "StDev Calc" contains standard deviation calculations between technical replicates and across biological replicates for all samples; TechRep is short for technical replicate, BR is short for biological replicate. Sheet titled "Error Calculation" contains propagated error calculations, which includes technical and biological error. Sheet titled "qPCR Analysis (Cell Type)" contains relative expression calculations for all samples; in this case, ESC samples are compared to WT H9 ESCs which are set to a relative value of 1 and neuron samples are compared to WT H9 neurons which are set to a relative value of 1. Sheet titled "Statistics (Cell Type)" contains all statistics calculations per gene based on calculations performed in "qPCR Analysis (Cell Type)." Sheet titled "For R (Cell Type)" is to be saved as a.csv file and input into R for plotting; this was used to generate Figs 2 and 3B–3D. Sheet titled "qPCR Analysis (to WT ESCs)" contains relative expression calculations for all samples; in this case, all samples are compared to WT H9 ESCs which are set to a relative value of 1. Sheet titled "Statistics (to WT ESCs)" contains all statistics calculations per gene based on calculations performed in "qPCR Analysis (to WT ESCs)." Sheet titled "For R (to WT ESCs)" is to be saved as a.csv file and input into R for plotting; this was used to generate S7 Fig.
(XLSX)

**S3 Table. Raw data from RT-qPCR screening clones for *UBE3A* expression.** Tfx1-4 stands for transfections 1 through 4. AG1-0 is the ASdel1-0 patient-derived iPSC line. H9 is wild type hESCs.
(XLSX)

**S4 Table. Data from copy number assay for UBE3A.** This data was generated by CopyCaller (v2.1) software.
(XLSX)

**S5 Table. Methylation assay data and analysis.** Sheet titled "Instructions" contains instructions for analyzing custom EpiTect Methyl II PCR Array results. Sheet titled "Gene Table" contains the PCR array name (*SNRPN*) and the well positions of the cell lines analyzed. Sheet titled "Raw Data" contains the inputted CT values obtained for each digestion for all cell lines. Sheet titled "QC Data Report" contains the auto-calculated failure rate to be used for quality control measures. Sheet titled "Results" contains the auto-calculated percent methylation for each cell line; UM is unmethylated, M is methylated. Sheet titled "Calculations" contains the formulas for the calculations performed. Sheet titled "Summary Raw Data" summarizes the CT values for all digestions for all cell lines.
(XLSX)

## Acknowledgments

This study makes use of data generated by the DECIPHER community. A full list of centres who contributed to the generation of the data is available from https://deciphergenomics.org/about/stats and via email from contact@deciphergenomics.org. DECIPHER is hosted by EMBL-EBI and funding for the DECIPHER project was provided by the Wellcome Trust [grant number WT223718/Z/21/Z]. Those who carried out the original analysis and collection of DECIPHER data bear no responsibility for the further analysis or interpretation of the data. We thank Yaling Liu from the University of Connecticut Cell and Genome Engineering Core for her help with expansion and storage of stem cell lines used in this study. We thank Dr. Judy Brown and Lisa LaBelle from the Center for Genome Innovation at the University of Connecticut Institute for Systems Genomics for their help in performing CytoSNP analysis.

## Author Contributions

**Conceptualization:** Stormy J. Chamberlain.

**Data curation:** Rachel B. Gilmore.

**Formal analysis:** Rachel B. Gilmore, Dea Gorka.

**Funding acquisition:** Justin Cotney, Stormy J. Chamberlain.

**Investigation:** Rachel B. Gilmore, Dea Gorka, Pooja Sonawane.

**Project administration:** Justin Cotney, Stormy J. Chamberlain.

**Resources:** Rachel B. Gilmore, Dea Gorka, Christopher E. Stoddard.

**Supervision:** Justin Cotney, Stormy J. Chamberlain.

**Visualization:** Dea Gorka.

**Writing – original draft:** Rachel B. Gilmore, Dea Gorka.

**Writing – review & editing:** Rachel B. Gilmore, Dea Gorka, Christopher E. Stoddard, Pooja Sonawane, Justin Cotney, Stormy J. Chamberlain.

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
