## [Decision Letter · Decision Letter 0]

23 Jan 2024

PONE-D-23-32075Generation of isogenic models of Angelman syndrome and Prader-Willi syndrome in CRISPR/Cas9-engineered human embryonic stem cellsPLOS ONE

Dear Dr. Cotney,

Thank you for submitting your manuscript to PLOS ONE. After careful consideration, we feel that it has merit but does not fully meet PLOS ONE’s publication criteria as it currently stands. Therefore, we invite you to submit a revised version of the manuscript that addresses the points raised during the review process.

In fact several comments where pointing that better characterization and replication of experiments are needed. 

We look forward to receiving your revised manuscript.

Kind regards,

Osman El-Maarri, Ph.D

Academic Editor

PLOS ONE

Journal Requirements:

3. Thank you for stating the following in the Acknowledgments Section of your manuscript: "This work was supported by National Institutes of Health (NIH) grant T32HG010463 (R.B.G.), R35GM119465 (J.C.), R01 HD099975 (J.C. and S.J.C.), and R01HD094953 (S.J.C.). This study makes use of data generated by the DECIPHER community. A full list of centres who contributed to the generation of the data is available from https://deciphergenomics.org/about/stats and via email from contact@deciphergenomics.org. DECIPHER is hosted by EMBL-EBI and funding for the DECIPHER project was provided by the Wellcome Trust [grant number WT223718/Z/21/Z]. We thank Yaling Liu from the University of Connecticut Cell and Genome Engineering Core for her help with expansion and storage of stem cell lines used in this study. We thank Dr. Judy Brown and Lisa LaBelle from the Center for Genome Innovation at the University of Connecticut Institute for Systems Genomics for their help in performing CytoSNP analysis."

Please remove any funding-related text from the manuscript and let us know how you would like to update your Funding Statement. Currently, your Funding Statement reads as follows: "National Institutes of Health (NIH) grants: T32HG010463 (R.B.G.), R35GM119465 (J.C.), R01 HD099975 (J.C. and S.J.C.), and R01HD094953 (S.J.C.) https://www.nih.gov/

The funders did not play any role in study design, data collection and analysis, decision to publish, or preparation of the manuscript."

4. Thank you for stating the following in the Competing Interests section: "S.J.C. is now an employee of F. Hoffmann-La Roche AG.".

Reviewers' comments:

Reviewer's Responses to Questions

**Comments to the Author**

1. Is the manuscript technically sound, and do the data support the conclusions?

Reviewer #1: Partly

Reviewer #2: Yes

Reviewer #3: Yes

Reviewer #4: Partly

2. Has the statistical analysis been performed appropriately and rigorously? 

Reviewer #1: No

Reviewer #2: Yes

Reviewer #3: N/A

Reviewer #4: No

3. Have the authors made all data underlying the findings in their manuscript fully available?

Reviewer #1: Yes

Reviewer #2: Yes

Reviewer #3: Yes

Reviewer #4: Yes

4. Is the manuscript presented in an intelligible fashion and written in standard English?

Reviewer #1: Yes

Reviewer #2: Yes

Reviewer #3: Yes

Reviewer #4: Yes

5. Review Comments to the Author

Reviewer #1: This manuscript describes the production of new isogenic ES cell lines for modelling Angelman syndrome and Prader Willi syndrome. These ES cell lines will be very useful for the community studying these syndromes. I found the manuscript easy to read and appreciate the work that has gone into generating these lines.

My main concern is that the claims made about the qRT-PCR is not well-replicated or analysed in a statistically rigorous fashion. For the most part there aren't replicates shown on the graphs, but rather just one replicate per cell line. Without replicates, here this would be samples of RNA from different times for the same cell line, statistical analyses are not possible and therefore the reader is left to interpret the qRT-PCR that has small fold changes without the benefit of knowing the underlying variation, and therefore are left to just go with the interpretation of the authors. As it stands, reductions of 50% in pluripotency factor expression is not viewed as meaningful, whereas the effect on imprinted gene expression of 50% is (because it is expected). So some further statistics and replication would be very important.

I would like to see that all qRT-PCR analyses are replicated i.e. increase the replicate numbers to a minimum of 3, and then statistical analyses are performed.

I am concerned that the control H9 cells may just be the parental line, rather than having gone through the clonal cell line stress similar to the edited samples. This is probably necessary considering that they see reduction in several pluripotency factors measured in Fig 2A in all edited lines. Perhaps this is a side effect of clonal line generation.

I would also be interested to know the interpretation of why UBE3A-ATS is down in the mat deleted neurons. This transcript is on the paternal allele so should be retained.

As a more minor point, the discussion is highly repetitive with the results. I think this could be trimmed quite a lot so that the results are discussed rather than repeated. In this case, given that the manuscript focuses on creation of new reagents, it would likely be quite a short discussion.

Reviewer #2: This is a very well written manuscript on the generation of isogenic hESC lines as tools to study Angelman syndrome and Prader Willi syndrome. The work appears scientifically sound and will provide a valuable resource to the community.

I have one major comment:

It does not make sense to me that only one line was chosen for neuronal differentiation. Given that only four lines have been established and were characterized, all four should be subjected to all experiments, including the neuronal differentiation.

I have a few minor comments:

Line 43: Hypopigmentation is not a symptom of AS per se. Only in large deletion cases, involving the OCA2 gene, this is the case.

Line 47: PWS is not considered a "multigenic" disorder, but rather a "contiguous gene disorder".

Line 49: The potential contribution of the MAGEL2 gene (and related Schaaf-Yang syndrome) should also be included.

Reviewer #3: In this study, Gilmore et al. report the generation of isogenic hESC models of Angelman syndrome and Prader-Willi syndrome using CRISPR/Cas9 technology. The authors transfected H9 ES cells with a plasmid encoding CRISPR/Cas9 and gRNA targeting GOLGA8 and screened for puromycin resistant colonies. The survival colonies were further screeded for the reduction of UBE3A expression and gene copy number at about 50% as inducers for single allelic genomic deletion at the 15q11-13 region. The parent of origin of the genomic deletion was determined by methylation analysis at the Prader-Willi Syndrome Imprinting Center on paternal chromosome. As the result, two colonies with maternal deletion and one colony with paternal deletion were identified. The approximate size of the deletion in each colony was further determined by CytoSNP analysis. The authors went on to evaluate the pluripotency of one hESC line and its potential to differentiate into neuronal cells. Overall, the authors report two isogenic hESC lines with megabase-scale deletion modeling Angelman syndrome and one for Prader-Willi syndrome. The existence of such hESC models should benefit the understanding of disease mechanisms of these disorders and future therapeutic discoveries. However, several concerns remain to be further addressed as listed below.

Major comments:

1. In this study, mutant cells were screened for puromycin resistance. However, it is unclear whether the positive cell lines are stable lines with Cas9 and puromycin expression. If so, the expression of Cas9 and gRNA may cause the deletion of the additional copy of genome, thus resulting in homozygous deletion in the same region in later time. Therefor it is important to demonstrate the three characterized mutant lines are free of Cas9 and gRNA expression.

2. Results reported in Figure 3 and 4, Supplemental Figure 4, were conducted with only one or two repeats. It is important to increase n to provide more accurate measurement and report proper statistical analysis.

3. The authors evaluate the neuronal differentiation of H9Δmat15q_1 ESCs (Figure 4A, B and Supplemental Figure 4). As shown in Figure 4B, the levels of UBE3A-ATS and Pax6 in differentiated H9Δmat15q_1 cells were lower than H9 control cells, suggesting a lower potential of H9Δmat15q_1 ESCs in differentiating into neuronal cells. This possibility should be further investigated. In addition, PAX6 was the only neuronal maker evaluated. In fact, PAX6 is frequently used as a marker for neural progenitor cells, but not differentiated mature neurons. I suggest the authors to quantitatively reevaluate the differentiation of H9Δmat15q_1 ESCs using markers for mature neurons by immunofluorescence staining.

Minor comments:

1. H9Δmat15q_1 and H9Δmat15q_2groups were shown in the same color. It will be helpful to distinguish them using different colors.

Reviewer #4: This manuscript describes the derivation and characterization of three new hESC models of Angelman and Prader-Willi syndromes isogenic to the well characterized H9 ESC line. These are models of the large deletions and are challenging to generate by CRISPR/Cas9 approaches but the authors were successful by screening a large number of clones for possible large deletions. I have a few suggestions to improve rigor.

1. Statistical tests should be performed for the results graphed in figures 2,3 and 4.

2. Some of the experiments appear to only include 1 replicate because there are no error bars. These analyses should be repeated, with error bars and statistics performed.

6. PLOS authors have the option to publish the peer review history of their article (what does this mean?). If published, this will include your full peer review and any attached files.

Reviewer #1: No

Reviewer #2: No

Reviewer #3: No

Reviewer #4: **Yes: **Janine LaSalle

---

## [Author Response · Author response to Decision Letter 0]

25 Jun 2024

AUTHOR RESPONSE TO REVIEWERS

We thank the reviewers for their helpful comments and constructive feedback, which has aided us in strengthening the rigor and clarity of the manuscript. While reviewers recognized the utility of these lines and their importance to the field, reviewers were generally concerned with the low number of biological replicates, lack of statistical analysis, and exclusion of a subset of the cell lines generated from neuronal differentiation. As such, we have included three biological replicates for qPCR analyses, included statistical analysis of qPCR results, and provided characterization of all cell lines as mature 10-week differentiated neurons. Please find our detailed response to all reviewer comments (in blue) below.

Reviewer's Responses to Questions

Comments to the Author

1. Is the manuscript technically sound, and do the data support the conclusions?

Reviewer #1: Partly

Reviewer #2: Yes

Reviewer #3: Yes

Reviewer #4: Partly

2. Has the statistical analysis been performed appropriately and rigorously? 

Reviewer #1: No

Reviewer #2: Yes

Reviewer #3: N/A

Reviewer #4: No

3. Have the authors made all data underlying the findings in their manuscript fully available?

Reviewer #1: Yes

Reviewer #2: Yes

Reviewer #3: Yes

Reviewer #4: Yes

4. Is the manuscript presented in an intelligible fashion and written in standard English?

Reviewer #1: Yes

Reviewer #2: Yes

Reviewer #3: Yes

Reviewer #4: Yes

5. Review Comments to the Author

Reviewer #1: This manuscript describes the production of new isogenic ES cell lines for modelling Angelman syndrome and Prader Willi syndrome. These ES cell lines will be very useful for the community studying these syndromes. I found the manuscript easy to read and appreciate the work that has gone into generating these lines.

My main concern is that the claims made about the qRT-PCR is not well-replicated or analysed in a statistically rigorous fashion. For the most part there aren't replicates shown on the graphs, but rather just one replicate per cell line. Without replicates, here this would be samples of RNA from different times for the same cell line, statistical analyses are not possible and therefore the reader is left to interpret the qRT-PCR that has small fold changes without the benefit of knowing the underlying variation, and therefore are left to just go with the interpretation of the authors. As it stands, reductions of 50% in pluripotency factor expression is not viewed as meaningful, whereas the effect on imprinted gene expression of 50% is (because it is expected). So some further statistics and replication would be very important.

I would like to see that all qRT-PCR analyses are replicated i.e. increase the replicate numbers to a minimum of 3, and then statistical analyses are performed.

All samples have now been analyzed in three biological replicates and two-three technical replicates. For statistical analysis, a one way ANOVA with Dunnett’s test has been performed.

I am concerned that the control H9 cells may just be the parental line, rather than having gone through the clonal cell line stress similar to the edited samples. This is probably necessary considering that they see reduction in several pluripotency factors measured in Fig 2A in all edited lines. Perhaps this is a side effect of clonal line generation.

We believe this comment is actually in reference to Fig 3A. After repeating the experiment with three biological replicates, we observe only one statistically significant reduction of a pluripotency gene, SOX2. This is observed in the two Angelman-like lines. Whether this is biological or an artifact of this system would require further examination. As most pluripotency genes show no significant difference between the edited lines and the parental H9 line, we don’t believe the genome editing had a drastic effect on the pluripotency status of these lines. We would anticipate further systematic characterization of these lines to support this hypothesis based on the observed morphology, behavior, and ability to differentiate into the neuronal lineage of all edited lines.

I would also be interested to know the interpretation of why UBE3A-ATS is down in the mat deleted neurons. This transcript is on the paternal allele so should be retained.

We do not observe a statistically significant decrease in UBE3A-ATS in maternal deletion lines after analyzing additional biological replicates and performing statistical analysis (Figure 4C and S5 Fig).

As a more minor point, the discussion is highly repetitive with the results. I think this could be trimmed quite a lot so that the results are discussed rather than repeated. In this case, given that the manuscript focuses on creation of new reagents, it would likely be quite a short discussion.

The discussion was trimmed to remove repetitive sections, while still focusing on points not clarified in the results section.

Reviewer #2: This is a very well written manuscript on the generation of isogenic hESC lines as tools to study Angelman syndrome and Prader Willi syndrome. The work appears scientifically sound and will provide a valuable resource to the community.

I have one major comment:

It does not make sense to me that only one line was chosen for neuronal differentiation. Given that only four lines have been established and were characterized, all four should be subjected to all experiments, including the neuronal differentiation.

All cell lines described have now been included in neuronal differentiation and characterization. We have only included one of the Angelman-like lines, H9Δmat15q11-q13_1, in the main figure (Fig 4) for a few reasons. The deletion harbored by the H9Δmat15q11-q13_1 is a more common deletion found in AS patients compared to the deletion in the H9Δmat15q11-q13_2 line. Additionally, for one of the biological replicates of the H9Δmat15q11-q13_2 line we obtained very low yield of RNA precluding reliable qPCR measurements for one biological replicate. We wanted to be sure this line was included in the characterization and publication but admit further replication is likely warranted, and thus we chose to move this cell line to the supplement (S5 Fig) for neuronal characterization.

I have a few minor comments:

Line 43: Hypopigmentation is not a symptom of AS per se. Only in large deletion cases, involving the OCA2 gene, this is the case.

This point has been clarified.

Line 47: PWS is not considered a "multigenic" disorder, but rather a "contiguous gene disorder".

The wording has been changed.

Line 49: The potential contribution of the MAGEL2 gene (and related Schaaf-Yang syndrome) should also be included.

This point has been included.

Reviewer #3: In this study, Gilmore et al. report the generation of isogenic hESC models of Angelman syndrome and Prader-Willi syndrome using CRISPR/Cas9 technology. The authors transfected H9 ES cells with a plasmid encoding CRISPR/Cas9 and gRNA targeting GOLGA8 and screened for puromycin resistant colonies. The survival colonies were further screeded for the reduction of UBE3A expression and gene copy number at about 50% as inducers for single allelic genomic deletion at the 15q11-13 region. The parent of origin of the genomic deletion was determined by methylation analysis at the Prader-Willi Syndrome Imprinting Center on paternal chromosome. As the result, two colonies with maternal deletion and one colony with paternal deletion were identified. The approximate size of the deletion in each colony was further determined by CytoSNP analysis. The authors went on to evaluate the pluripotency of one hESC line and its potential to differentiate into neuronal cells. Overall, the authors report two isogenic hESC lines with megabase-scale deletion modeling Angelman syndrome and one for Prader-Willi syndrome. The existence of such hESC models should benefit the understanding of disease mechanisms of these disorders and future therapeutic discoveries. However, several concerns remain to be further addressed as listed below.

Major comments:

1. In this study, mutant cells were screened for puromycin resistance. However, it is unclear whether the positive cell lines are stable lines with Cas9 and puromycin expression. If so, the expression of Cas9 and gRNA may cause the deletion of the additional copy of genome, thus resulting in homozygous deletion in the same region in later time. Therefor it is important to demonstrate the three characterized mutant lines are free of Cas9 and gRNA expression.

We apologize for the lack of clarity in describing the transfection and selection. The Cas9 cassette is not stably integrated in these lines; both Cas9 and puromycin resistance are expressed transiently. We added clarification to the methods section to better explain this process.

2. Results reported in Figure 3 and 4, Supplemental Figure 4, were conducted with only one or two repeats. It is important to increase n to provide more accurate measurement and report proper statistical analysis.

Three biological replicates have been included for all lines and statistical analysis has been performed.

3. The authors evaluate the neuronal differentiation of H9Δmat15q_1 ESCs (Figure 4A, B and Supplemental Figure 4). As shown in Figure 4B, the levels of UBE3A-ATS and Pax6 in differentiated H9Δmat15q_1 cells were lower than H9 control cells, suggesting a lower potential of H9Δmat15q_1 ESCs in differentiating into neuronal cells. This possibility should be further investigated. In addition, PAX6 was the only neuronal maker evaluated. In fact, PAX6 is frequently used as a marker for neural progenitor cells, but not differentiated mature neurons. I suggest the authors to quantitatively reevaluate the differentiation of H9Δmat15q_1 ESCs using markers for mature neurons by immunofluorescence staining.

We have now included data for neuronal differentiation of all lines (Fig 4). While the main figure only shows one of the AS-like lines, H9Δmat15q11-q13_1, an equivalent characterization of the other AS-like line, H9Δmat15q11-q13_2, is included in S5 Fig. In addition, we increased the number of neuronal genes measured by including DLG4, the gene which encodes PSD-95 protein, and MAP2 to our analysis. For clarity, we moved the neuronal genes, including PAX6, to their own graph. We also performed immunocytochemistry for NeuN & MAP2 proteins (S5 Fig), though believe that immunofluorescence experiments should only be used as a qualitative measure and thus have not quantified them in any way.

Minor comments:

1. H9Δmat15q_1 and H9Δmat15q_2groups were shown in the same color. It will be helpful to distinguish them using different colors.

We modified our color scheme so each cell line is distinguished by its own color throughout the manuscript.

Reviewer #4: This manuscript describes the derivation and characterization of three new hESC models of Angelman and Prader-Willi syndromes isogenic to the well characterized H9 ESC line. These are models of the large deletions and are challenging to generate by CRISPR/Cas9 approaches but the authors were successful by screening a large number of clones for possible large deletions. I have a few suggestions to improve rigor.

1. Statistical tests should be performed for the results graphed in figures 2,3 and 4.

As Fig 2 describes the screening process, it would be inappropriate to perform statistical analysis. The results of the screen were followed by more rigorous characterization. Additional biological replicates were added, and statistical analysis was performed on data presented in Figs 3 and 4.

2. Some of the experiments appear to only include 1 replicate because there are no error bars. These analyses should be repeated, with error bars and statistics performed.

Additional biological replicates were added, error bars have been included on relevant graphs, and statistical analysis has been performed on data presented in Fig 3, Fig 4, S4, S5, and S6.

6. PLOS authors have the option to publish the peer review history of their article (what does this mean?). If published, this will include your full peer review and any attached files.

Do you want your identity to be public for this peer review? For information about this choice, including consent withdrawal, please see our Privacy Policy.

Reviewer #1: No

Reviewer #2: No

Reviewer #3: No

Reviewer #4: Yes: Janine LaSalle

Manuscript and file formatting has been updated accordingly.

Cell lines are available through the UConn Cell and Genome Engineering Core upon request and completion of MTA. Raw data from qPCR and analysis is provided in S2 Table. Code used for generation of graphs is publicly available on Github, as listed in the Methods section.

3. Thank you for stating the following in the Acknowledgments Section of your manuscript: "This work was supported by National Institutes

---

## [Decision Letter · Decision Letter 1]

23 Jul 2024

PONE-D-23-32075R1Generation of isogenic models of Angelman syndrome and Prader-Willi syndrome in CRISPR/Cas9-engineered human embryonic stem cellsPLOS ONE

Dear Dr. Cotney,

Thank you for submitting your manuscript to PLOS ONE. After careful consideration, we feel that it has merit but does not fully meet PLOS ONE’s publication criteria as it currently stands. Therefore, we invite you to submit a revised version of the manuscript that addresses the points raised during the review process. Before we move to final acceptance you need as authors to comments and satisfy one of the reviwers who asked for minor revision.

We look forward to receiving your revised manuscript.

Kind regards,

Osman El-Maarri, Ph.D

Academic Editor

PLOS ONE

Journal Requirements:

Reviewers' comments:

Reviewer's Responses to Questions

**Comments to the Author**

1. If the authors have adequately addressed your comments raised in a previous round of review and you feel that this manuscript is now acceptable for publication, you may indicate that here to bypass the “Comments to the Author” section, enter your conflict of interest statement in the “Confidential to Editor” section, and submit your "Accept" recommendation.

Reviewer #1: All comments have been addressed

Reviewer #2: All comments have been addressed

Reviewer #3: (No Response)

2. Is the manuscript technically sound, and do the data support the conclusions?

Reviewer #1: (No Response)

Reviewer #2: Yes

Reviewer #3: No

3. Has the statistical analysis been performed appropriately and rigorously? 

Reviewer #1: (No Response)

Reviewer #2: Yes

Reviewer #3: Yes

4. Have the authors made all data underlying the findings in their manuscript fully available?

Reviewer #1: (No Response)

Reviewer #2: Yes

Reviewer #3: Yes

5. Is the manuscript presented in an intelligible fashion and written in standard English?

Reviewer #1: (No Response)

Reviewer #2: Yes

Reviewer #3: Yes

6. Review Comments to the Author

Reviewer #1: (No Response)

Reviewer #2: The authors have done a good job addressing the reviewers' comments. I do not have any further comments.

Reviewer #3: The authors did not adequately address our only main concern. One (easy) experiment is necessary:

The cell lines the authors made are likely to be an important resource for the Angelman syndrome research community. Some in the field may seek to use these cell lines to test genome editor-based therapies. The authors used plasmids encoding Cas9, the guide RNAs, and puromycin resistance. If Cas9 and/or guide RNAs are stably integrated into the genome of these cells, this has the potential to confound future experiments that use the same Cas variant and/or different guide RNAs and could lead to instability of the lines themselves.

Circular plasmids, when transiently transfected, can stably integrate into the mammalian genome. A recent study nicely quantifies this phenomenon: Lim et al (https://pubmed.ncbi.nlm.nih.gov/37100816/).

The authors attempted to address our main concern by writing the following in the methods section:

“As the Cas9 cassette was not stably integrated into these cell lines, selection was continued for 48 hours total to select cells transiently expressing the vector containing the gRNA, Cas9 protein, and puromycin resistance”

However, they provide no data nor performed any experiments to confirm that the Cas9 cassette was not stably integrated.

The authors already have genomic DNA and RNA isolated from these cells. We are thus requesting that the authors perform PCR (with genomic DNA) and RT-PCR (with RNA) to confirm that Cas9, the guide RNAs, and puromycin sequences are not present in the genome or expressed in the cell lines.

7. PLOS authors have the option to publish the peer review history of their article (what does this mean?). If published, this will include your full peer review and any attached files.

Reviewer #1: No

Reviewer #2: No

Reviewer #3: No

---

## [Decision Letter · Decision Letter 2]

23 Sep 2024

Generation of isogenic models of Angelman syndrome and Prader-Willi syndrome in CRISPR/Cas9-engineered human embryonic stem cells

PONE-D-23-32075R2

Dear Dr. Cotney,

We’re pleased to inform you that your manuscript has been judged scientifically suitable for publication and will be formally accepted for publication once it meets all outstanding technical requirements.

Kind regards,

Osman El-Maarri, Ph.D

Academic Editor

PLOS ONE

Additional Editor Comments (optional):

Reviewers' comments:

Reviewer's Responses to Questions

**Comments to the Author**

1. If the authors have adequately addressed your comments raised in a previous round of review and you feel that this manuscript is now acceptable for publication, you may indicate that here to bypass the “Comments to the Author” section, enter your conflict of interest statement in the “Confidential to Editor” section, and submit your "Accept" recommendation.

Reviewer #3: All comments have been addressed

2. Is the manuscript technically sound, and do the data support the conclusions?

Reviewer #3: Yes

3. Has the statistical analysis been performed appropriately and rigorously? 

Reviewer #3: Yes

4. Have the authors made all data underlying the findings in their manuscript fully available?

Reviewer #3: Yes

5. Is the manuscript presented in an intelligible fashion and written in standard English?

Reviewer #3: Yes

6. Review Comments to the Author

Reviewer #3: The remaining minor concern has been adequately addressed.

7. PLOS authors have the option to publish the peer review history of their article (what does this mean?). If published, this will include your full peer review and any attached files.

Reviewer #3: No

---

## [Editor Report · Acceptance letter]

25 Oct 2024

PONE-D-23-32075R2 

PLOS ONE

Dear Dr. Cotney, 

I'm pleased to inform you that your manuscript has been deemed suitable for publication in PLOS ONE. Congratulations! Your manuscript is now being handed over to our production team.

Kind regards, 

on behalf of

Priv.-Doz. Dr. Osman El-Maarri 

Academic Editor

PLOS ONE